# Impact of PARP inhibitor maintenance therapy in newly diagnosed advanced epithelial ovarian cancer: A meta-analysis

**Banghyun Lee[1], Suk-Joon Chang [2]\*, Byung Su Kwon[3], Joo-Hyuk Son[2], Myong Cheol Lim[4], Yun Hwan Kim [5], Shin-Wha Lee[6], Chel Hun Choi[7], Kyung Jin Eoh[8], Jung-Yun Lee[9], Dong Hoon Suh[10], Yong Beom Kim[10]**

1 Department of Obstetrics and Gynecology, Inha University Hospital, Inha University College of Medicine, Incheon, Republic of Korea, 2 Department of Obstetrics and Gynecology, Ajou University School of Medicine, Suwon-si, Gyeonggi-do, Republic of Korea, 3 Department of Obstetrics and Gynecology, School of Medicine, Kyung Hee University Medical Center, Kyung Hee University, Seoul, Republic of Korea, 4 Center for Gynecologic Cancer, Research Institute and Hospital, National Cancer Center, Goyang-si, Gyeonggi-do, Republic of Korea, 5 Department of Obstetrics and Gynecology, Ewha Womans University Mokdong Hospital, Ewha Womans University College of Medicine, Seoul, Republic of Korea, 6 Department of Obstetrics and Gynecology, University of Ulsan College of Medicine, Asan Medical Center, Seoul, Republic of Korea, 7 Department of Obstetrics and Gynecology, Samsung Medical Center, Sungkyunkwan University School of Medicine, Seoul, Republic of Korea, 8 Department of Obstetrics and Gynecology, Yongin Severance Hospital, Yonsei University College of Medicine, Yongin-si, Gyeonggi-do, Republic of Korea, 9 Department of Obstetrics and Gynecology, Yonsei University College of Medicine, Seoul, Republic of Korea, 10 Department of Obstetrics and Gynecology, Seoul National University Bundang Hospital, Seongnam-si, Gyeonggi-do, Republic of Korea

\* drchang@ajou.ac.kr

**Data Availability Statement:** All relevant data are within the paper and its Supporting Information files.

## Abstract

### Objectives

This meta-analysis was undertaken to systematically evaluate the effects of poly (ADP-ribose) polymerase inhibitor (PARPi) maintenance therapy on the survival of newly diagnosed advanced epithelial ovarian cancer (EOC) patients.

### Methods/Materials

A systematic literature search revealed 3,227 studies. A subsequent selection process identified seven suitable randomized studies that assessed the survival outcomes in newly diagnosed advanced EOC patients administered PARPi (n = 1921; the PARPi group) or placebo (n = 1150; the placebo group). The survival outcomes were compared with respect to the PARPi treatment regardless of bevacizumab maintenance therapy. All adverse events ≥ grade 3 were analyzed. Review Manager Version 5.4.1 software was used for the meta-analysis.

### Results

The two-year progression-free survival (PFS) was significantly better in the PARPi group than the placebo (Hazard ratio [HR], 0.53; 95% confidence interval [CI], 0.41 to 0.68). Furthermore, patients in the PARPi group with the *BRCA1/2* mutation (BRCAm), BRCA wild

**Funding:** The author(s) received no specific funding for this work.

**Competing interests:** The authors have declared that no competing interests exist.

type, homologous-recombination deficiency (HRD), or HRD without BRCAm, but not with homologous-recombination proficiency had a significantly better two-year PFS than the patients in the placebo group. The five-year overall survival (OS) was comparable in the two groups, but patients in the PARPi group with BRCAm had a significantly better five-year OS than those in the placebo group (HR, 0.57; 95% CI, 0.44 to 0.74). In addition, the adverse event rate ($\geq$ grade 3) was significantly higher in the PARPi group than in the placebo group (HR, 2.94; 95% CI, 1.13 to 7.63).

## Conclusions

In patients with newly diagnosed advanced EOC, PARPi maintenance therapy was significantly more effective in terms of survival than no PARPi treatment. However, the risk of serious adverse events was higher for patients who received PARPi maintenance therapy.

## Introduction

Ovarian cancer is the leading cause of death from gynecologic cancer [1], with epithelial ovarian cancer (EOC) accounting for 90% of ovarian cancers [2]. EOC patients usually present with advanced-stage disease at diagnosis [2,3]. Although the response rate to combined cytoreductive surgery and platinum-based chemotherapy is high, approximately 80% of patients experience recurrence [4,5]. Unfortunately, the treatment for recurrence becomes ineffective with repeated recurrent episodes [6].

High-grade serous carcinoma (HGSC) accounts for $\sim$70% of EOC cases; 13% of HGSC patients have a germline mutation in *BRCA1/2*, and $\sim$50% have a somatic homologous recombination deficiency (HRD) [7]. Poly (adenosine diphosphate [ADP]–ribose) polymerases (PARPs) are a family of proteins involved in various cellular processes, including DNA repair [8], and poly (ADP-ribose) polymerase inhibitors (PARPis) play important roles in the treatment of cancers deficient in double-strand break repair, which includes HRDs [9,10]. Furthermore, *BRCA1/2* proteins help repair double-strand DNA breaks through homologous recombination [11], and in tumors with a *BRCA1/2* mutation (BRCAm). PARPis prevent efficient double-strand break repair, which causes cell death [12], and the repair of DNA damage to cancer cells caused by cytotoxic chemotherapy [13].

Previous randomized controlled trials (RCTs) reported that PARPis significantly improve progression-free survival (PFS) when used as maintenance therapy in platinum-sensitive recurrent EOC patients regardless of the status of biomarkers like BRCAm or HRD [14–17]. Recently, several RCTs have shown that PARPi maintenance therapy significantly improves PFS in newly diagnosed advanced EOC patients with or without BRCAm or HRD [18–22]. Moreover, two recent RCTs reported the survival benefit of PARPis on the overall survival (OS) of newly diagnosed advanced EOC patients with BRCAm or HRD [23,24].

In newly diagnosed advanced EOC patients, several therapeutic strategies such as platinum-based chemotherapy with or without bevacizumab, dose-dense platinum-based chemotherapy and intraperitoneal chemotherapy have been reported to have less than satisfactory effects on survival [25–28]. On the other hand, many studies have focused on improving survival in newly diagnosed advanced EOC patients. In particular, several recent RCTs have reported the effects of PARPis on survival in newly diagnosed advanced EOC patients dependent on the expression of BRCAm or HRD [18–24]. Therefore, this study examined the impact

of PARPis on PFS, OS, and adverse events of newly diagnosed advanced EOC patients with or without BRCAm or HRD through a systematic review and meta-analysis of these RCTs.

## Materials and methods

This systematic review and meta-analysis was conducted based on the Cochrane Handbook for Systematic Reviews of Interventions throughout the entire process [29]. On the other hand, a specific protocol does not exist. A completed PRISMA (Preferred Reporting Items for Systematic Reviews and Meta-Analyses) checklist and flow diagram were provided.

### Search strategy

The PubMed, Cochrane Library, Embase, and KoreaMed databases were searched for all relevant studies in October 2022 using the following combination of keywords: (ovarian cancer OR tubal cancer OR peritoneal cancer) AND (PARPi OR Niraparib OR Rucaparib OR Olaparib OR Veliparib OR Talazoparib OR Iniparib OR Pamiparib OR Lynparza OR Rubraca OR Zejula OR Talzenna) AND (survival OR mortality OR death) (S1 Table). Additional studies considered relevant but not identified by the database searches were identified by examining the references in the selected clinical studies and review articles.

### Selection criteria

The inclusion criteria were studies that examined the following: histologically diagnosed EOC, newly diagnosed advanced EOC with responses after first-line platinum-taxane chemotherapy, use of PARPis, placebo control, and survival. The exclusion criteria were as follows: non-RCTs, review articles, editorials, letters, protocols, clinical responses, and irrelevant studies such as laboratory articles. In studies that included overlapping groups of patients, only those with the most comprehensive data were included in the meta-analysis to avoid duplicate information.

### Data extraction, outcomes of interest, and risk bias

Two investigators independently extracted the data of interest from studies using a checklist. Discrepancies between investigators were resolved by discussion. The eligible population was dichotomized into patients administered a PARPi or placebo. The data retrieved from studies included the study name, authors, year of publication, study design, number of patients (intention-to-treat (ITT) population), treatment arms, duration of maintenance, median age, histologic type, median follow-up duration, median PFS, median overall survival, survival rate, adverse event $\geq$ grade 3, and number of patients with a BRCAm, BRCA wild type, HRD, HRD without BRCAm, or homologous-recombination proficiency (HRP).

The primary outcome variable was the PFS, which is defined as the time between randomization and disease progression or death from any cause. OS was defined as the time between randomization and death from any cause. The common Terminology Criteria for Adverse Events (CTCAE) v5.0 was used to evaluate adverse events [30].

The qualities of the included studies were critically appraised separately by two investigators using the revised Cochrane risk of bias tool for randomized trials (RoB 2.0 version) [31].

### Evaluation of the overall quality of evidence

The quality of evidence for the outcomes was evaluated using the guidelines of the Grading of Recommendations Assessment, Development, and Evaluation (GRADE) system. These guidelines involved the sequential assessment of evidence quality, evaluation of risk-benefit balance,

and subsequent appraisal of the strengths of recommendations [32]. The qualities of evidence were reported as follows: high quality, indicating that further research is unlikely to change the confidence in the estimation of effect; moderate quality, indicating that further research is likely to have an important impact on confidence in the estimate of effect and may change the estimation; low quality, indicating further research is highly likely to an important impact on confidence in the estimate of effect and is likely to change the estimation; very low quality, indicating little confidence in the estimate of the effect.

### Statistical analysis

This study investigated whether PARPi reduced the risks of ovarian cancer progression, death, and adverse events. Random-effects models were implemented with the Inverse Variance method to analyze the survival and the Mantel–Haenszel method to analyze the adverse events. The hazard ratios (HRs) for survival and the odds ratios (ORs) for adverse events and their 95% confidence intervals (CIs) were calculated. The $I^2$ statistic and Cochran's Q statistic, which are heterogeneity indices, were used to determine if there was a dispersion among HRs and ORs across the studies assessed. Survivals were analyzed in the overall population and subgroups. GRADE evidence profiles were produced using GRADEpro GDT. The meta-analysis was conducted using Review Manager Version 5.4.1 software (The Nordic Cochrane Centre, Copenhagen, Denmark), and P values of <0.05 were deemed significant.

## Results

### Search results and study characteristics

The literature search identified 3,227 potentially relevant studies, but only seven investigations on five RCTs that met the selection criteria were eventually included (Fig 1).

Table 1 lists the characteristics of these studies. Four trials [18–20,22,23] included the overall population except for one trial [21,24], which included patients with BRCAm alone. PARPi was used as maintenance therapy after chemotherapy in three trials [18,21,22,24], used concurrently with chemotherapy and then as maintenance therapy in one trial [19], and used in addition to maintenance bevacizumab after bevacizumab/chemotherapy in one trial [20,23]. Three trials [18,19,22] reported the PFS alone, whereas the two trials [20,21,23,24] reported the PFS and OS. Five papers [18–21,24] were full-text articles, and two [22,23] were abstracts. Five studies [18–22] were used to evaluate the PFS: two [23,24] to evaluate the OS and four [18–20,24] to evaluate adverse events ≥ grade 3.

The risk of bias assessments for each study revealed low to unclear risk in five domains (S1 Fig).

### Two-year PFS–comparison between the PARPi and placebo groups

In the overall population, the PFS was significantly better in the PARPi group than in the placebo group (HR, 0.53; 95% CI, 0.41 to 0.68; P <0.00001; $I^2$, 84%; 3071 patients; moderate quality evidence). Moreover, for patients with BRCAm, BRCA wild type, HRD, or HRD without BRCAm, the PFS was significantly improved by PARPi treatment (BRCAm: HR, 0.35; 95% CI, 0.29 to 0.42; P <0.00001; $I^2$, 0%; 1049 patients; high-quality evidence) (BRCA wild type: HR, 0.75; 95% CI, 0.65 to 0.88; P = 0.0003; $I^2$, 0%; 1068 patients; high-quality evidence) (HRD: HR, 0.44; 95% CI, 0.32 to 0.60; P <0.00001; $I^2$, 69%; 1181 patients; high-quality evidence) (HRD without BRCAm: HR, 0.58; 95% CI, 0.38 to 0.89; P = 0.01; $I^2$, 74%; unknown patient number; high-quality evidence). On the other hand, PFSs were comparable in the PARPi and placebo

**PRISMA 2020 flow diagram for new systematic reviews which included searches of databases and registers only**

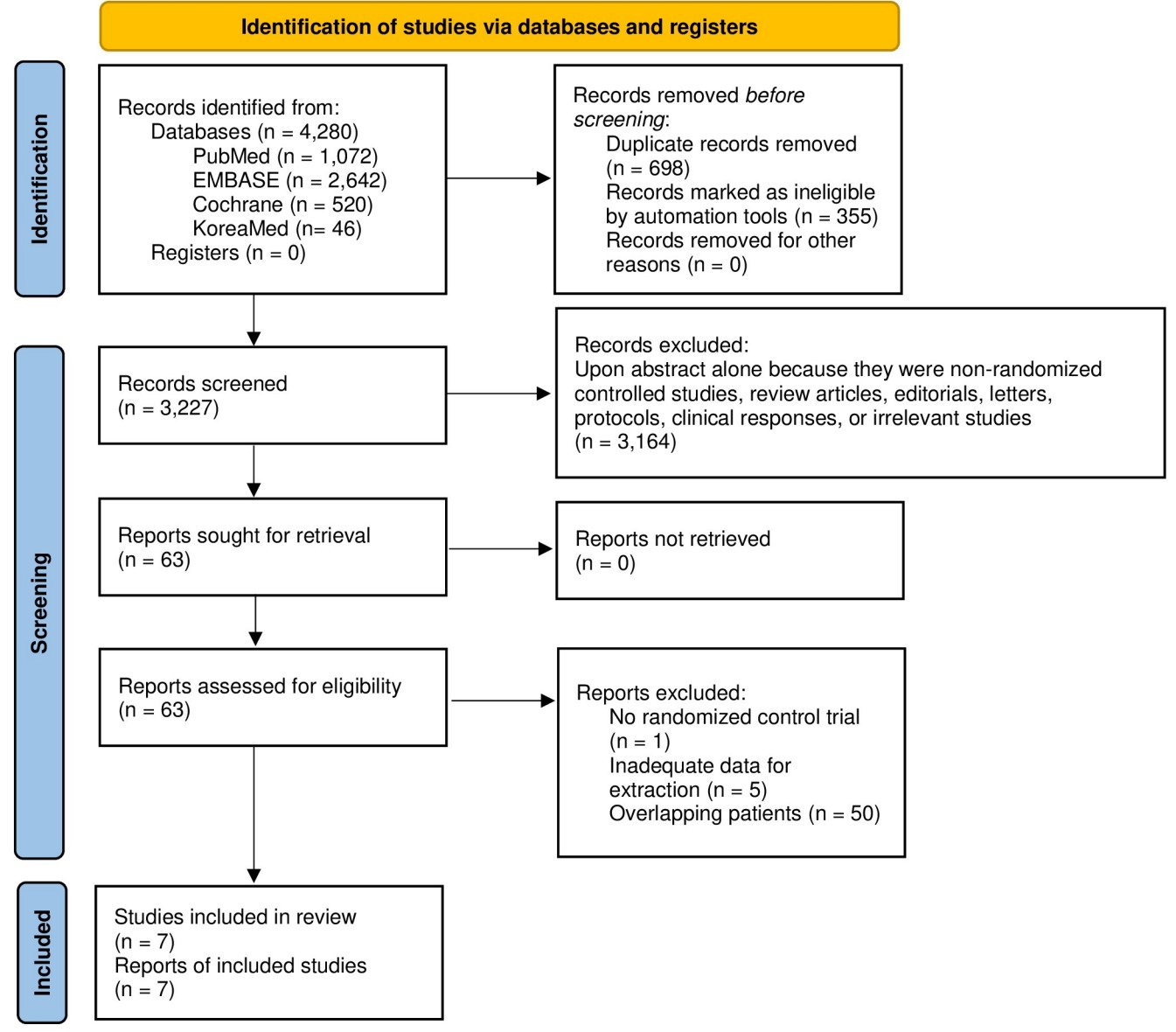

**Fig 1. Flow chart of study selection.**

groups for patients with HRP (HR, 0.83; 95% CI, 0.66 to 1.03; $P$ = 0.09; I$^2$, 36%; 775 patients; moderate quality evidence) (Fig 2 and S2 Table).

## Five-year OS–comparison between the PARPi and placebo groups

In the overall population, the OSs were similar in the PARPi and placebo groups (HR, 0.73; 95% CI, 0.44 to 1.20; $P$ = 0.21; I$^2$, 86%; 1197 patients; low-quality evidence), but the OS was significantly better for patients with BRCAm in the PARPi group (HR, 0.57; 95% CI, 0.44 to

**Table 1. Characteristics of the included studies.**

| Trial | Author, Year | Study Design | Number of participants (ITT population) | Treatment arms | Median age (range) (y) | Histologic types | Median duration of follow up (mo) | Median PFS (mo) | Median OS (mo) | Survival rate | Adverse events (grade ≥3) |
|---|---|---|---|---|---|---|---|---|---|---|---|
| PRIMA | Gonzalez-Martin et al. [18] 2019 | RCT, Phase 3 | PARPi: 487 Placebo: 246 | Niraparib versus Placebo, Dosage: 300 mg orally once daily | PARPi: 62 (32–85) Placebo: 62 (33–88) | Serous type: 94.8% | 13.8 | PARPi: 13.8 Placebo: 8.2 | | All population: PARPi: 232/487 (47.6%) Placebo: 155/246 (63.0%) BRCA mutation: PARPi: 49/152 (32.2%) Placebo: 40/71 (56.3%) HRD without BRCA mutation: PARPi: 32/95 (33.7%) Placebo: 33/55 (60.0%) HRP: PARPi: 111/169 (65.7%) Placebo: 56/80 (70.0%) | PARPi: 341/484 (70.5%) (most common: anemia, thrombocytopenia, neutropenia, platelet count decreased, fatigue) Placebo: 46/244 (18.9%) |
| VELIA | Coleman et al. [19] 2019 | RCT, Phase 3 | PARPi: 382 Placebo: 375 | Veliparib versus Placebo, Dosage: 400 mg orally twice daily | PARPi:62 (30–85) Placebo: 62 (33–86) | High-grade serous type: 100% | 28 | PARPi: 23.5 Placebo: 17.3 | | All population: PARPi: 191/382 (50%) Placebo: 237/375 (63.2%) BRCA mutation: PARPi: 34/108 (31.5%) Placebo: 49/90 (54.4%) BRCA wild type: PARPi: 142/245 (58%) Placebo: 171/254 (67.3%) HRD: PARPi: 87/214 (40.7%) Placebo: 124/207 (59.9%) HRP: PARPi: 80/125 (64%) Placebo: 89/124 (71.8%) | PARPi: 332/377 (88%) (most common: neutropenia, anemia, thrombocytopenia, leukopenia, fatigue, nausea) Placebo: 285/371 (77%) |

*(Continued)*

**Table 1.** (Continued)

| Trial | Author, Year | Study Design | Number of participants (ITT population) | Treatment arms | Median age (range) (y) | Histologic types | Median duration of follow up (mo) | Median PFS (mo) | Median OS (mo) | Survival rate | Adverse events (grade ≥3) |
|---|---|---|---|---|---|---|---|---|---|---|---|
| PAOLA-1 | Ray-Coquard et al. [20] 2019 | RCT, Phase 3 | PARPi: 537 Placebo: 269 | Olaparib + Bevacizumab versus Placebo + Bevacizumab, Dosage:300 mg orally twice daily | PARPi: 61 (32–87) Placebo: 60 (26–85) | Serous type: 95.8% | PARPi: 22.7 Placebo: 24.0 | PARPi: 22.1 Placebo: 16.6 | | All population: PARPi: 280/537 (52%) Placebo: 194/269 (72%) BRCA mutation: PARPi: 41/157 (26%) Placebo: 49/80 (61%) BRCA wild type: PARPi: 239/380 (63%) Placebo: 145/189 (77%) HRD: PARPi: 87/255 (34%) Placebo: 92/132 (70%) HRP: PARPi: 145/192 (76%) Placebo: 66/85 (78%) | PARPi: 303/535 (57%) (most common: hypertension, anemia, lymphopenia, neutropenia, fatigue) Placebo: 136/267 (51%) |
| | Ray-Coquard et al. [23] 2022 | RCT, Phase 3 (Abstract) | PARPi: 537 Placebo: 269 | Olaparib + Bevacizumab versus Placebo + Bevacizumab, Dosage:300 mg orally twice daily | PARPi: 61 (32–87) Placebo: 60 (26–85) | Serous type: 95.8% | PARPi: 61.7 Placebo: 61.9 | | PARPi: 56.5 Placebo: 51.6 | All population: PARPi: 288/537 (53.6%) Placebo: 158/269 (58.7%) BRCA mutation: PARPi: 80/37 (30.6%) Placebo: 157/48 (46.3%) HRD: PARPi: 93/255 (36.5%) Placebo: 69/132 (52.3%) | |
| SOLO1 | Banerjee et al. [21] 2021 | RCT, Phase 3 | PARPi: 260 Placebo: 131 | Olaparib versus Placebo, Dosage:300 mg orally twice daily | PARPi: 53 (29–82) Placebo: 53 (31–84) | High-grade serous type: 96.2% | PARPi: 57.6 Placebo: 60 | PARPi: 56.0 Placebo: 13.8 | | All population and BRCA mutation: PARPi: 118/260 (45%) Placebo: 100/131 (76%) | PARPi: 55/260 (21%) (most common: anemia, neutropenia, fatigue) Placebo: 17/130 (13%) |
| | DiSilvestro et al. [24] 2022 | RCT, Phase 3 | PARPi: 260 Placebo: 131 | Olaparib versus Placebo, Dosage:300 mg orally twice daily | PARPi: 53 (29–82) Placebo: 53 (31–84) | High-grade serous type: 96.2% | PARPi: 88.9 Placebo: 87.4 | PARPi: not reached Placebo: 75.2 | | All population and BRCA mutation: PARPi: 84/260 (32.3%) Placebo: 65/131 (49.6%) | PARPi: 103/260 (39.6%) (most common: anemia, neutropenia, fatigue) Placebo: 26/130 (20.0%) |
| PRIME | Li et al. [22] 2022 | RCT, Phase 3 (Abstract) | PARPi: 255 Placebo: 129 | Niraparib versus Placebo, Dosage: 300 mg or 200mg orally once daily | | | 27.5 | PARPi: 24.8 Placebo: 8.3 | | | PARPi versus Placebo: anemia (18.0% versus 1.6%), neutrophil count decreased (17.3% versus 1.6%), and platelet count decreased (14.1% versus 0.8%) |

HRD, homologous-recombination deficiency; HRP, homologous-recombination proficiency; ITT, intention-to-treat; mo, month; OS, overall survival; PARPi, poly (adenosine diphosphate [ADP]–ribose) polymerase inhibitors PFS, progression-free survival; RCT, randomized controlled studies; y, year.

**A. Overall population**

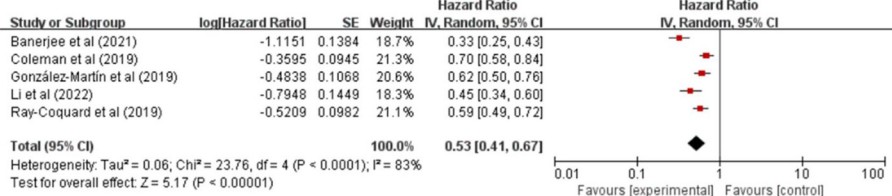

**B. Patients with BRCAm**

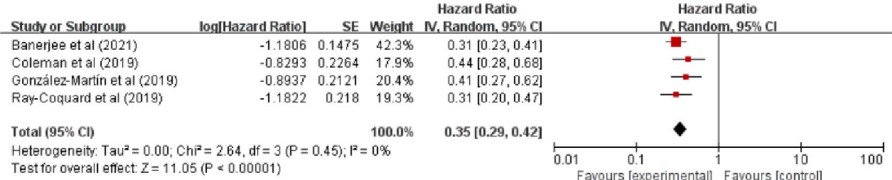

**C. Patients with BRCA wild type**

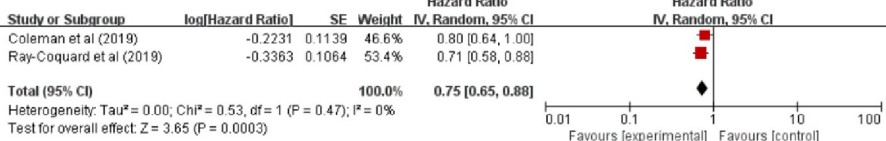

**D. Patients with HRD**

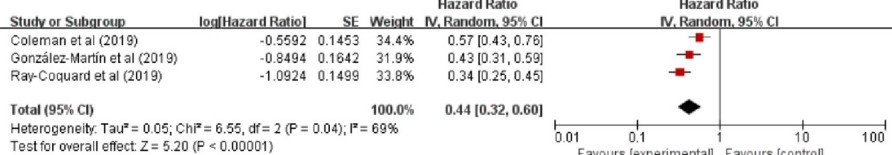

**E. Patients with HRD without BRCAm**

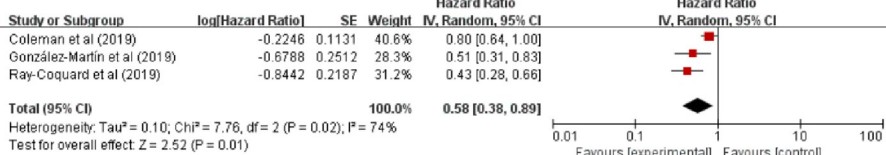

**F. Patients with HRP**

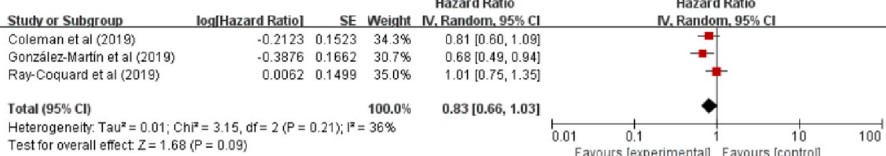

**Fig 2. Two-year PFS.**

## A. Overall population

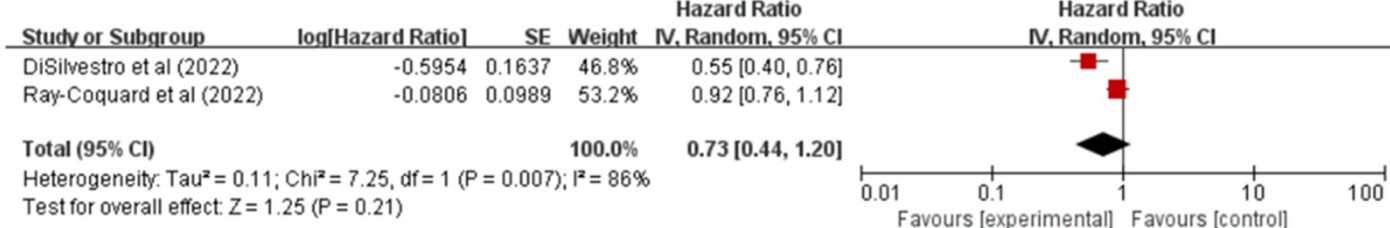

## B. Patients with BRCAm

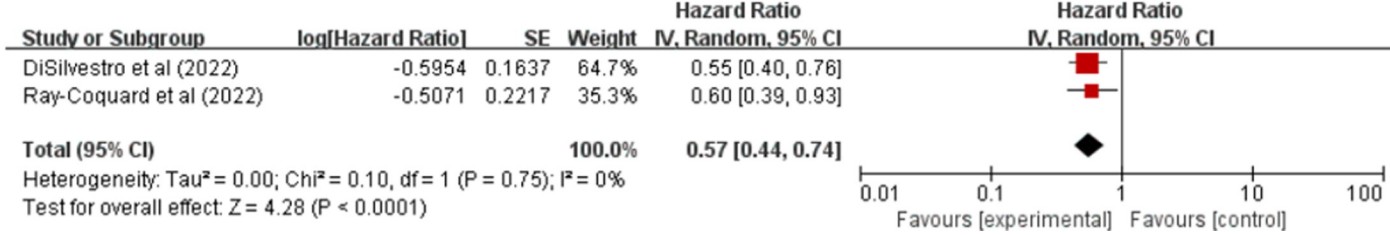

**Fig 3. Five-year OS.**

0.74; $P < 0.0001$; $I^2$, 0%; 628 patients; high-quality evidence) than the placebo group (Fig 3 and S2 Table).

### Adverse events–comparison between the PARPi and placebo groups

Table 1 lists the most common adverse events (grade ≥3) in each study. In the overall population, the rate of adverse events of ≥ grade 3 was significantly higher in the PARPi group (OR, 2.94; 95% CI, 1.13 to 7.63; $P = 0.03$; $I^2$, 96%; 2668 patients; moderate-quality evidence) than the placebo group (Fig 4 and S2 Table).

## Discussion

The present meta-analysis was conducted to determine the effects and safety of PARPi maintenance therapy on the survival of patients with newly diagnosed advanced EOC. PARPis improved PFS significantly in the overall population and patients with BRCAm, BRCA wild

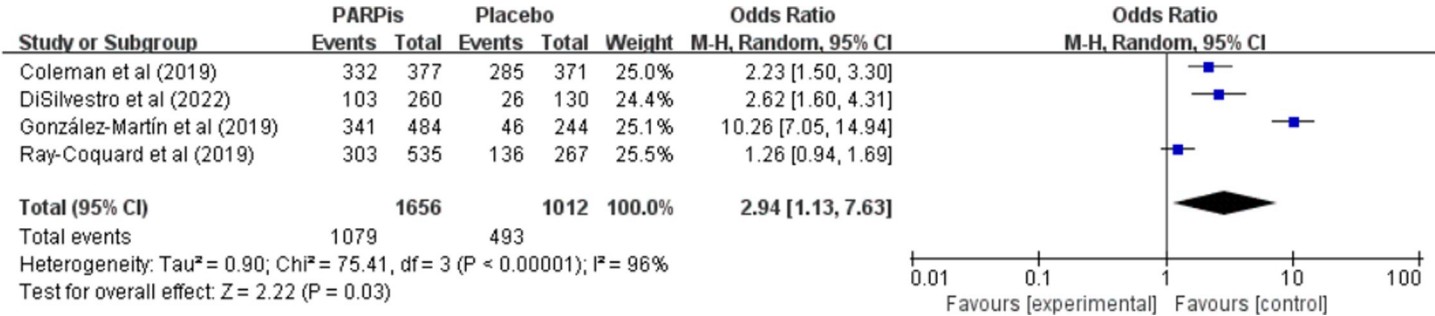

**Fig 4. Adverse events ≥ grade 3.** Meta-analysis was performed on the overall population.

type, HRD, or HRD without BRCAm but did not influence PFS in patients with HRP. Furthermore, the PARPi treatment improved the OS significantly in patients with BRCAm but did not influence the OS in the overall population. In addition, the PARPi treatment was associated with a significantly higher rate of adverse events of $\geq$ grade 3 in the overall population.

In patients with newly diagnosed advanced EOC, PARP inhibitors improve the PFS in stage III, stage Iv, complete response and partial response to first-line chemotherapy, and patients with and without residual macroscopic disease after debulking surgery [33]. Recently, five RCTs concluded that PARPi treatment significantly improved survival, regardless of the biomarker status, when administered as maintenance therapy to newly diagnosed advanced EOC patients with a complete or partial response to platinum-based chemotherapy. In the PRIMA trial, niraparib improved the PFS significantly compared to the placebo when used as maintenance therapy after platinum-based chemotherapy in the overall population and in patients with BRCAm, HRD, HRD without BRCAm, or HRP, but not in patients with BRCA wild type [18]. In the VELIA trial, veliparib improved the PFS significantly compared to platinum-based chemotherapy plus a placebo when co-administered with platinum-based chemotherapy and subsequently as maintenance therapy in the overall patients and patients with BRCAm or HRD but not in patients with the BRCA wild type, HRD without BRCAm, or HRP [19]. In the PAOLA-1 trial, the addition of olaparib to maintenance bevacizumab after bevacizumab/platinum-based chemotherapy improved the PFS significantly in the overall population and patients with BRCAm, BRCA wild type, HRD, or HRD without BRCAm, but not HRP, and improved the OS in patients with BRCAm or HRD (HR 0.62), but not in the overall population and patients with HRD without BRCAm or HRP (abstract only) [20,23]. In the SOLO1 trial, olaparib improved the PFS and OS significantly compared to the placebo when used as maintenance therapy after platinum-based chemotherapy in patients with BRCAm (overall population) [21,24]. Finally, in the PRIME trial, niraparib significantly improved the PFS compared to the placebo when used as maintenance therapy after platinum-based chemotherapy for the overall patients and patients with HRD (HR 0.48) or HRP (HR 0.41) (abstract only) [22].

The meta-analysis performed on an ITT basis showed that PARPi improved the PFS significantly but not the OS in the overall population and PFS and OS in patients with BRCAm. These OS results might be due to the small number of studies because only two were analyzed [23,24]. Nevertheless, the analysis shows that the PARPi treatment improved PFS and OS significantly in newly diagnosed advanced EOC patients with BRCAm [21,24].

Interestingly, five studies included in the PFS analyses revealed significant or marginally significant HRs for an improved PFS for the overall PARPi-treated patients and PARPi-treated patients with BRCAm, BRCA wild type, HRD, or HRD without BRCAm. These findings support the results of the meta-analyses, namely, that PARPi significantly improved the PFS in newly diagnosed advanced EOC patients. Regarding patients with HRP, the PRIMA trial showed that PARPi treatment improved PFS significantly, whereas the VELIA and PAOLA-1 trials found PARPi-treatment resulted in insignificant HRs [18–20]. The meta-analysis showed that the PARPi-treatment did not influence the PFS in newly diagnosed advanced EOC with HRP, highlighting the need for HRD testing to guide PARPi maintenance decision-making in advanced EOC.

Furthermore, the analysis showed that anemia, thrombocytopenia, neutropenia, fatigue, and nausea were common adverse events (grade $\geq$ 3) of a PARPi treatment [18–22,24]. In the PAOLA-1 trial, the addition of maintenance olaparib to bevacizumab/platinum-based chemotherapy did not increase the incidence of serious adverse events [20], but in the PRIMA, VELIA, and SOLO1 trials, PARPi administration increased the risk of adverse events (grade $\geq$ 3) significantly [18,19,21,24]. Similarly, the meta-analysis showed that PARPi

significantly increased the risk of adverse events ≥ grade 3 when administered to newly diagnosed advanced EOC patients.

The strength of this meta-analysis is that systematic analysis was performed in all currently available randomized studies. This meta-analysis added one most recent study for PFS analysis compared with previous meta-analyses [33–35]. Moreover, to the best of the authors' knowledge, this meta-analysis is the first to evaluate systematically the effect of PARPi treatment on OS of newly diagnosed advanced EOC patients. Therefore, these results can help guide the treatment of newly diagnosed advanced EOC patients.

Nevertheless, this study had some limitations. First, only seven studies in the analysis could be included because only five RCTs provided adequate data. On the other hand, most meta-analyses results provided moderate or high-quality evidence. Second, the RCTs included were conducted using different types of PARP inhibitors with different characteristics. Previous studies showed that these different inhibitors have similar effects on the survival outcomes in ovarian cancer patients [14–17]. Moreover, the risk of adverse events (grade ≥ 3) of different PARP inhibitors was evaluated in previous meta-analyses [34–36]. Third, one study that included bevacizumab in the case and control regimens was included in the analyses of the PFS, OS, and adverse events [20,23]. Unfortunately, the effects of bevacizumab could not be evaluated because of the few eligible studies.

## Conclusions

The present systematic review and meta-analysis showed that PARPi maintenance therapy significantly improved the PFS and OS but increased the incidence of serious adverse events in newly diagnosed advanced EOC patients. The small number of studies limited the significance of this meta-analysis. On the other hand, these results provide clinically useful information on the impact of PARPi maintenance therapy. Therefore, large-scale multicenter RCTs will be needed to confirm the present findings regarding the effects of PARPi maintenance therapy on survival, especially the OS, of newly diagnosed advanced EOC patients.

## Supporting information

**S1 Checklist. PRISMA 2020 checklist.**
(DOCX)

**S1 Fig. Risk of bias.** (A) two-year PFS, (B) five-year OS, and (C) Adverse events ≥ grade 3.
(PDF)

**S1 Table. Search strategy.** (A) Pubmed, (B) Cochrane Library, (C) Embase, and (D) KoreaMed.
(PDF)

**S2 Table. GRADE evidence profiles.** CI: confidence interval; HR: hazard ratio; HRD: homologous-recombination deficiency; HRP: homologous-recombination proficiency; OR: odds ratio; OS: overall survival; PARPi: poly (adenosine diphosphate [ADP]–ribose) polymerase inhibitors; PFS: progression-free survival; RCT: randomized controlled studies. a. High heterogeneity unexplained, b. Wide confidence interval that crosses 1 and is out of the area between 0.75 and 1.25.
(PDF)

## Author Contributions

**Conceptualization:** Banghyun Lee, Suk-Joon Chang, Byung Su Kwon, Joo-Hyuk Son, Myong Cheol Lim, Yun Hwan Kim, Shin-Wha Lee, Chel Hun Choi, Kyung Jin Eoh, Jung-Yun Lee, Dong Hoon Suh, Yong Beom Kim.

**Data curation:** Banghyun Lee, Byung Su Kwon.

**Formal analysis:** Banghyun Lee, Suk-Joon Chang.

**Methodology:** Banghyun Lee, Suk-Joon Chang.

**Project administration:** Suk-Joon Chang.

**Resources:** Banghyun Lee, Joo-Hyuk Son.

**Visualization:** Banghyun Lee.

**Writing – original draft:** Banghyun Lee, Suk-Joon Chang.

**Writing – review & editing:** Banghyun Lee, Suk-Joon Chang, Byung Su Kwon, Joo-Hyuk Son, Myong Cheol Lim, Yun Hwan Kim, Shin-Wha Lee, Chel Hun Choi, Kyung Jin Eoh, Jung-Yun Lee, Dong Hoon Suh, Yong Beom Kim.

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
