## [Decision Letter · Decision Letter 0]

2 Oct 2023

PONE-D-23-26174Impact ofPARP inhibitor maintenance therapy in newly diagnosed advanced epithelial ovarian cancer: A meta-analysisPLOS ONE

Dear Dr. Chang,

Thank you for submitting your manuscript to PLOS ONE. After careful consideration, we feel that it has merit but does not fully meet PLOS ONE’s publication criteria as it currently stands. Therefore, we invite you to submit a revised version of the manuscript that addresses the points raised during the review process.

The manuscript was reviewed by at least two independent reviewers and their review comments are appended below. The reviewers identified several concerns that should be addressed by the authors during the revision. Should you addressed these comments fully the manuscript will be considered for publication.

We look forward to receiving your revised manuscript.

Kind regards,

Asmerom Tesfamariam Sengal, MD, PhD

Academic Editor

PLOS ONE

2. Please amend either the title on the online submission form (via Edit Submission) or the title in the manuscript so that they are identical.

Reviewers' comments:

Reviewer's Responses to Questions

**Comments to the Author**

1. Is the manuscript technically sound, and do the data support the conclusions?

Reviewer #1: Yes

Reviewer #2: Partly

2. Has the statistical analysis been performed appropriately and rigorously? 

Reviewer #1: Yes

Reviewer #2: No

3. Have the authors made all data underlying the findings in their manuscript fully available?

Reviewer #1: Yes

Reviewer #2: Yes

4. Is the manuscript presented in an intelligible fashion and written in standard English?

Reviewer #1: Yes

Reviewer #2: No

5. Review Comments to the Author

Reviewer #1: The manuscript entitled “Impact ofPARP inhibitor maintenance therapy in newly diagnosed advanced epithelial ovarian cancer: A meta-analysis” by Lee et al. presents a comprehensive metaanalysis of studies investigating the effects of PARPi maintenance therapy in advanced EOC. The paper provides insights into the effect of PARPi therapy in advanced. There are a couple of points need to be addressed:

My major critique is that the study did not use a systematic review protocol, which may potentially introduce bias and affect the overall reliability of the results. This has been acknowledged in the paper, however, the reason behind it was not clarified. Another major limitation is the limited studies included for the meta-analysis.

In addition, there are a couple of minor points to consider:

1. Better to label the subgroups in each panel of Fig2 and Fig3.

2. It would be good to include a brief explanation of the I2 statistic and Cochran's Q statistics used to assess the heterogeneities as they are not common tests.

Reviewer #2: Chang et al. conducted a systematic review and meta-analysis on “Impact of PARP inhibitor maintenance therapy in newly diagnosed advanced epithelial ovarian cancer (EOC)”. They eventually inferred that in newly diagnosed advanced EOC patients, PARPi maintenance therapy was significantly more effective in terms of survival compared to placebo group. However, there are major issues without addressing which the manuscript will not go to publication in PLOS ONE. Below are point-by-point comments and concerns that warrant clarity and/or corrections.

Abstract:

Title: Please put a space after “of”

Objectives:

Please keep the statement as below, because evaluation of adverse effects was not the main aim of this study.

“This meta-analysis was undertaken to systematically evaluate the effects of poly (ADP-ribose) polymerase inhibitor (PARPi) maintenance therapy on survival of newly diagnosed advanced epithelial ovarian cancer (EOC) patients.”

Methods:

Please include the software used for statistical analysis as this study is a meta-analysis.

Results:

Hazard ratio [HR] for disease progression or death, and confidence interval [CI] to be initially written in full and abbreviations can be applied thereafter.

Introduction:

1. Overall the introduction is shallow and authors are recommended to clearly mention key concepts of the study, figure out why the topic is important, provide literature review related to this topic, identify existing gaps and finally define the aim of the study.

2. In lines 112 through 113 “In newly diagnosed advanced EOC patients, therapeutic strategies have been reported to have less than satisfactory effects on survival”. Would you please elaborate the therapeutic strategies?

Materials and methods:

1. Kindly clarify the statement “In this meta-analysis, a protocol for systematic review was not used”?

2. The timeframe of identified published studies and their geographical coverage with regard to participants of the RCTs selected for analysis?

3. Please mention the specific guideline applied to conduct this systematic review and meta-analysis and include relevant reference for the same? For instance, PRISMA or else.

Selection criteria:

Please clarify what you meant by “irrelevant studies”? Non-English publications were included or not? If not, please specify it in that way.

Kindly re-write the whole paragraph using medically sound and complex language. It is not necessary to repeat the same word several times. For example, the word “studies” has been used in every point of the inclusion criteria.

Data Extraction:

What guideline was used to extract data related to the parameters e.g (name, authors, year of publication, study design, number of patients etc.)

Quality assessment:

Kindly clarify what kind of tool was applied to evaluate the domain of risk of bias for the included RCTs?

Please use “Statistical Analysis”, not “Statistical analyses” in the respective sub-heading.

In result section:

1. Provide detailed description of background information/study characteristics regarding the included RCTs. The figure alone is not enough.

2. The subtitles should be short and well descriptive. “Search results and characteristics and assessments of the risk of bias of the included studies”. Instead you can put “study characteristics” but remember to include detailed description of the included RCTs.

3. Table: The table is too huge and less organized.

4. The data for OS in the table in most of the rows are empty?

5. For SOLO1, the spaces are filled as “same” and no clarification is given for the item/data “same”.

6. Figures (forest plots) 2 through 4 are not clear, please provide a clear version of these figures.

Adverse effects:

Regarding the adverse effects, please mention the details of specific adverse effects of interest?

What guideline was followed while evaluating AEs, e.g (Common Terminology Criteria for Adverse Events: CTCAE) and include relevant reference for the same.

Also, considering that adverse effects vary across various types of PARPis, how could you perform meta-analysis for this outcome and hence provide inference?

Conclusion:

1. Kindly clarify whether the effect of PARP inhibitors on ovarian cancer is affected by FIGO stage status, response to first-line chemotherapy, and residual macroscopic disease after de-bulking/cyto-reduction surgery?

2. Line 241, the last portion of the statement does not seem complete.

3. Line 245-249, please clarify the paragraph and include relevant references.

4. In discussion part, authors should explain the implication of the findings to the current practice.

5. Kindly consider the below recent articles on the same topic and include them in your references.

• https://pubmed.ncbi.nlm.nih.gov/32654312// DOI: 10.1111/1471-0528.16411

• https://pubmed.ncbi.nlm.nih.gov/34727316// DOI: 10.1007/s12325-021-01959-5

• https://doi.org/10.1007/s00404-021-06070-2

• https://pubmed.ncbi.nlm.nih.gov/37217940// DOI: 10.1186/s12957-023-03027-4

• https://pubmed.ncbi.nlm.nih.gov/35354431// DOI: 10.1186/s12885-022-09455-x

6. PLOS authors have the option to publish the peer review history of their article (what does this mean?). If published, this will include your full peer review and any attached files.

Reviewer #1: No

Reviewer #2: **Yes: **Embaye Kidane Siele

---

## [Author Response · Author response to Decision Letter 0]

25 Oct 2023

Reviewer #1: The manuscript entitled “Impact ofPARP inhibitor maintenance therapy in newly diagnosed advanced epithelial ovarian cancer: A meta-analysis” by Lee et al. presents a comprehensive metaanalysis of studies investigating the effects of PARPi maintenance therapy in advanced EOC. The paper provides insights into the effect of PARPi therapy in advanced. There are a couple of points need to be addressed:

My major critique is that the study did not use a systematic review protocol, which may potentially introduce bias and affect the overall reliability of the results. This has been acknowledged in the paper, however, the reason behind it was not clarified.

The sentence was changed as follows: This systematic review and meta-analysis was conducted based on the Cochrane Handbook for Systematic Reviews of Interventions throughout the entire process [29]. On the other hand, a specific protocol does not exist. A completed PRISMA (Preferred Reporting Items for Systematic Reviews and Meta-Analyses) checklist and flow diagram were provided.

Reference 29 was added.

Another major limitation is the limited studies included for the meta-analysis.

We thoughts RCTs can provide more adequate data compared with observational studies. Therefore, we conducted meta-analysis including only RCTs.

Some previous studies have conducted meta-analysis using only RCTs as follows:

[33] Cheng H, Yang J, Liu H, Xiang Y. Poly (adenosine diphosphate [ADP]-ribose) polymerase (PARP) inhibitors as maintenance therapy in women with newly diagnosed ovarian cancer: a systematic review and meta-analysis. Arch Gynecol Obstet. 2021; 304: 285-296.

[34] Lin Q, Liu W, Xu S, Shang H, Li J, Guo Y, et al. PARP inhibitors as maintenance therapy in newly diagnosed advanced ovarian cancer: a meta-analysis. BJOG. 2021; 128: 485-493.

[35] Gulia S, Kannan S, Ghosh J, Rath S, Maheshwari A, Gupta S. Maintenance therapy with a poly(ADP-ribose) polymerase inhibitor in patients with newly diagnosed advanced epithelial ovarian cancer: individual patient data and trial-level meta-analysis. ESMO Open. 2022; 7:100558.

[36] Suh YJ, Lee B, Kim K, Jeong Y, Choi HY, Hwang SO, et al. Bevacizumab versus PARP-inhibitors in women with newly diagnosed ovarian cancer: a network meta-analysis. BMC Cancer. 2022; 22: 346.

In addition, there are a couple of minor points to consider:

1. Better to label the subgroups in each panel of Fig2 and Fig3.

In Fig2 and Fig3, labels of subgroups were inserted. Therefore, we provided revised Fig2 and Fig3.

2. It would be good to include a brief explanation of the I2 statistic and Cochran's Q statistics used to assess the heterogeneities as they are not common tests.

The sentence was changed as follows: The I2 statistic and Cochran's Q statistic, which are heterogeneity indices, were used to determine if there was a dispersion among HRs and ORs across the studies assessed.

Reviewer #2: Chang et al. conducted a systematic review and meta-analysis on “Impact of PARP inhibitor maintenance therapy in newly diagnosed advanced epithelial ovarian cancer (EOC)”. They eventually inferred that in newly diagnosed advanced EOC patients, PARPi maintenance therapy was significantly more effective in terms of survival compared to placebo group. However, there are major issues without addressing which the manuscript will not go to publication in PLOS ONE. Below are point-by-point comments and concerns that warrant clarity and/or corrections.

Abstract:

Title: Please put a space after “of”

It was corrected.

Objectives:

Please keep the statement as below, because evaluation of adverse effects was not the main aim of this study.

“This meta-analysis was undertaken to systematically evaluate the effects of poly (ADP-ribose) polymerase inhibitor (PARPi) maintenance therapy on survival of newly diagnosed advanced epithelial ovarian cancer (EOC) patients.”

It was changed

Methods:

Please include the software used for statistical analysis as this study is a meta-analysis.

The following sentence was inserted: Review Manager Version 5.4.1 software was used for the meta-analysis.

Results:

Hazard ratio [HR] for disease progression or death, and confidence interval [CI] to be initially written in full and abbreviations can be applied thereafter.

Those were corrected as follows: (Hazard ratio [HR], 0.53; 95% confidence interval [CI], 0.41 to 0.68)

Introduction:

1. Overall the introduction is shallow and authors are recommended to clearly mention key concepts of the study, figure out why the topic is important, provide literature review related to this topic, identify existing gaps and finally define the aim of the study.

Introduction was rewritten according to recommendation of reviewer.

References 1 and 6 were added.

2. In lines 112 through 113 “In newly diagnosed advanced EOC patients, therapeutic strategies have been reported to have less than satisfactory effects on survival”. Would you please elaborate the therapeutic strategies?

The sentence was changed as follows: In newly diagnosed advanced EOC patients, several therapeutic strategies such as platinum-based chemotherapy with or without bevacizumab, dose-dense platinum-based chemotherapy and intraperitoneal chemotherapy have been reported to have less than satisfactory effects on survival [25-28].

Materials and methods:

1. Kindly clarify the statement “In this meta-analysis, a protocol for systematic review was not used”?

PLOS ONE guideline asks the followings: “Authors must also state in their “Methods” section whether a protocol exists for their systematic review, and if so, provide a copy of the protocol as supporting information and provide the registry number in the abstract.”

The sentence was changed as follows: This systematic review and meta-analysis was conducted based on the Cochrane Handbook for Systematic Reviews of Interventions throughout the entire process [29]. On the other hand, a specific protocol does not exist. A completed PRISMA (Preferred Reporting Items for Systematic Reviews and Meta-Analyses) checklist and flow diagram were provided.

Reference 29 was added.

2. The timeframe of identified published studies and their geographical coverage with regard to participants of the RCTs selected for analysis?

Identified published studies were provided according to time sequence (publication year). Timeframe such as median duration of follow up, and median PFS and OS were collected to improve understandings of the studies.

All included studies were conducted in many international countries. Therefore, geographical coverage of participants were not collected.

3. Please mention the specific guideline applied to conduct this systematic review and meta-analysis and include relevant reference for the same? For instance, PRISMA or else.

The following sentence was inserted: This systematic review and meta-analysis was conducted based on the Cochrane Handbook for Systematic Reviews of Interventions throughout the entire process [29].

Selection criteria:

Please clarify what you meant by “irrelevant studies”? Non-English publications were included or not? If not, please specify it in that way.

The following words were inserted into sentence:…. irrelevant studies such as laboratory articles.

Non-English publications were included.

Kindly re-write the whole paragraph using medically sound and complex language. It is not necessary to repeat the same word several times. For example, the word “studies” has been used in every point of the inclusion criteria.

The entire paragraph were rewritten as follows: The inclusion criteria were studies that examined the following: histologically diagnosed EOC, newly diagnosed advanced EOC with responses after first-line platinum-taxane chemotherapy, use of PARPis, placebo control, and survival. The exclusion criteria were as follows: non-RCTs, review articles, editorials, letters, protocols, clinical responses, and irrelevant studies such as laboratory articles. In studies that included overlapping groups of patients, only those with the most comprehensive data were included in the meta-analysis to avoid duplicate information.

Data Extraction:

What guideline was used to extract data related to the parameters e.g (name, authors, year of publication, study design, number of patients etc.)

Cochrane guideline was used.

The following sentence was inserted: This systematic review and meta-analysis was conducted based on the Cochrane Handbook for Systematic Reviews of Interventions throughout the entire process [29].

Quality assessment:

Kindly clarify what kind of tool was applied to evaluate the domain of risk of bias for the included RCTs?

It was already provided in lines 163-165 as follows: …….. using the revised Cochrane risk of bias tool for randomized trials (RoB 2.0 version) [31].

Please use “Statistical Analysis”, not “Statistical analyses” in the respective sub-heading.

It was changed

In result section:

1. Provide detailed description of background information/study characteristics regarding the included RCTs. The figure alone is not enough.

In results, we already provided ‘Table 1’ and ‘S1 Fig’ as follows: Table 1 lists the characteristics of these studies, and S1 Fig presents the results of risk of bias assessments for each study.

Moreover, in discussion (lines 249-269), we already explained the included RCTs as follows: Recently, five RCTs concluded that PARPi treatment… ………….

Additionally, according to recommendation of the reviewer, we inserted the following sentences in the results: Table 1 lists the characteristics of these studies. Four trials [18-20,22,23] included the overall population except for one trial [21,24], which included patients with BRCAm alone. PARPi was used as maintenance therapy after chemotherapy in three trials [18,21,22.24], used concurrently with chemotherapy and then as maintenance therapy in one trial [19], and used in addition to maintenance bevacizumab after bevacizumab/chemotherapy in one trial [20,23]. Three trials [18,19,22] reported the PFS alone, whereas the two trials [20,21,23,24] reported the PFS and OS. Five papers [18-21,24] were full-text articles, and two [22,23] were abstracts. Five studies [18-22] were used to evaluate the PFS: two [23,24] to evaluate the OS and four [18-20,24] to evaluate adverse events ≥ grade 3. 

The following sentence was changed: Risk of bias assessments for each study revealed low to unclear risk in five domains (S1 Fig).

2. The subtitles should be short and well descriptive. “Search results and characteristics and assessments of the risk of bias of the included studies”. Instead you can put “study characteristics” but remember to include detailed description of the included RCTs

It was changed as follows: Search results and study characteristics

Detailed description of the included RCTs are same with those in above no. 1.

3. Table: The table is too huge and less organized.

To reduce size of the table, we removed the column of ‘Duration of maintenance’

4. The data for OS in the table in most of the rows are empty?

In the column of ‘Median OS (mo)’, the rows of two studies which provided OS have data and the rows of the other five studies which did not provided OS are empty. In addition, in the column of ‘Median PFS (mo)’, the rows of five studies which provided PFS have data and the rows of the other two studies which did not provided PFS are empty.

5. For SOLO1, the spaces are filled as “same” and no clarification is given for the item/data “same”.

In the study of DiSilvestro et al (2022) of SOLO1, ‘same’ was replaced with data. Additionally, in the study of Ray‑Coquard et al (2022) of PAOLA-1, ‘same’ was replaced with data, too.

6. Figures (forest plots) 2 through 4 are not clear, please provide a clear version of these figures.

We already provided the best clear version (tiff) that we can do. Those were also managed in PACE of PLOS ONE (https://pacev2.apexcovantage.com/). We think resolution of those figures was decreased when those were converted to PDF for reviewers. To help improvement of resolution, we attached PPT versions.

Adverse effects:

Regarding the adverse effects, please mention the details of specific adverse effects of interest?

In table 1, the most common adverse events (grade ≥3) according to each study were already included.

In discussion, we already mentioned those as follows: Furthermore, the analysis showed that anemia, thrombocytopenia, neutropenia, fatigue, and nausea were common adverse events (grade ≥ 3) of a PARPi treatment [18-22,24]. In the PAOLA-1 trial, the addition of maintenance olaparib to bevacizumab/platinum-based chemotherapy did not increase the incidence of serious adverse events [20], but in the PRIMA, VELIA, and SOLO1 trials, PARPi administration increased the risk of adverse events (grade ≥ 3) significantly [18,19,21,24]. Similarly, the meta-analysis showed that PARPi significantly increased the risk of adverse events ≥ grade 3 when administered to newly diagnosed advanced EOC patients.

In the lines 230, the following sentence was inserted: Table 1 lists the most common adverse events (grade ≥3) in each study.

What guideline was followed while evaluating AEs, e.g (Common Terminology Criteria for Adverse Events: CTCAE) and include relevant reference for the same.

CTCAE was used.

The following sentence was inserted into lines 161-162: The common Terminology Criteria for Adverse Events (CTCAE) v5.0 was used to evaluate adverse events [30].

Reference 30 was added.

Also, considering that adverse effects vary across various types of PARPis, how could you perform meta-analysis for this outcome and hence provide inference?

We analyzed the rate of adverse events (grade ≥3) from the studies which used three types of PARPis. In my knowledge, this kind of analysis is commonly performed in meta-analysis.

In discussion, we already mentioned as follows: Furthermore, the analysis showed that anemia, thrombocytopenia, neutropenia, fatigue, and nausea were common adverse events (grade ≥ 3) of a PARPi treatment [18-22,24]. 

In the limitation, the following sentence was inserted: Second, the RCTs included were conducted using different types of PARP inhibitors with different characteristics. Previous studies showed that these different inhibitors have similar effects on the survival outcomes in ovarian cancer patients [14-17]. Moreover, the risk of adverse events (grade ≥ 3) of different PARP inhibitors was evaluated in previous meta-analyses [34-36].

Conclusion:

1. Kindly clarify whether the effect of PARP inhibitors on ovarian cancer is affected by FIGO stage status, response to first-line chemotherapy, and residual macroscopic disease after de-bulking/cyto-reduction surgery?

The following sentences were inserted into lines 246-248: In patients with newly diagnosed advanced EOC, PARP inhibitors improve the PFS in stage Ⅲ, stage Ⅳ, complete response and partial response to first-line chemotherapy, and patients with and without residual macroscopic disease after debulking surgery [33]. Recently, five RCTs concluded that PARPi treatment…………….

2. Line 241, the last portion of the statement does not seem complete.

It was changed as follows: ……… based chemotherapy in patients with BRCAm (overall population) [21,24].

3. Line 245-249, please clarify the paragraph and include relevant references.

The paragraph was clarified and references were inserted as follows: The meta-analysis performed on an ITT basis showed that PARPi improved the PFS significantly but not the OS in the overall population and PFS and OS in patients with BRCAm. These OS results might be due to the small number of studies because only two were analyzed [23,24]. Nevertheless, the analysis shows that the PARPi treatment improved PFS and OS significantly in newly diagnosed advanced EOC patients with BRCAm [21,24].

4. In discussion part, authors should explain the implication of the findings to the current practice.

The following sentences were inserted in lines 293-298: The strength of this meta-analysis is that systematic analysis was performed in all currently available randomized studies. This meta-analysis added one most recent study for PFS analysis compared with previous meta-analyses [33-35]. Moreover, to the best of the authors’ knowledge, this meta-analysis is the ﬁrst to evaluate systematically the effect of PARPi treatment on OS of newly diagnosed advanced EOC patients. Therefore, these results can help guide the treatment of newly diagnosed advanced EOC patients.

5. Kindly consider the below recent articles on the same topic and include them in your references.

• https://pubmed.ncbi.nlm.nih.gov/32654312// DOI: 10.1111/1471-0528.16411

It was inserted as reference 34.

• https://pubmed.ncbi.nlm.nih.gov/34727316//DOI: 10.1007/s12325-021-01959-5

• https://doi.org/10.1007/s00404-021-06070-2

It was inserted as reference 33.

• https://pubmed.ncbi.nlm.nih.gov/37217940//DOI: 10.1186/s12957-023-03027-4

• https://pubmed.ncbi.nlm.nih.gov/35354431// DOI: 10.1186/s12885-022-09455-x

It was inserted as reference 36.

Additional changes

1. In table 1, HRD data (survival rate) of Ray‑Coquard et al. [2022] (PAOLA-1 trial) were inserted as follows: HRD: PARPi: 93/255 (36.5%), Placebo: 69/132 (52.3%).

2. In line 232, HR was replaced with OR as follows: ………. the PARPi group (OR, 2.94; 95% CI……….

3. In 238, ‘the safety and effect’ was changed with ‘the effects and safety’.

4. Reference 35 was added.

5. Number and format of references were changed.

6. The English in this document was rechecked by the expert who are skilled authors of English language papers (https://www.nurisco.net/).

---

## [Decision Letter · Decision Letter 1]

7 Nov 2023

Impact of PARP inhibitor maintenance therapy in newly diagnosed advanced epithelial ovarian cancer: A meta-analysis

PONE-D-23-26174R1

Dear Dr Chang ,

We’re pleased to inform you that your manuscript has been judged scientifically suitable for publication and will be formally accepted for publication once it meets all outstanding technical requirements.

Kind regards,

Asmerom Tesfamariam Sengal, MD, PhD

Academic Editor

PLOS ONE

Additional Editor Comments (optional):

Reviewers' comments:

Reviewer's Responses to Questions

**Comments to the Author**

1. If the authors have adequately addressed your comments raised in a previous round of review and you feel that this manuscript is now acceptable for publication, you may indicate that here to bypass the “Comments to the Author” section, enter your conflict of interest statement in the “Confidential to Editor” section, and submit your "Accept" recommendation.

Reviewer #1: All comments have been addressed

Reviewer #2: All comments have been addressed

2. Is the manuscript technically sound, and do the data support the conclusions?

Reviewer #1: Yes

Reviewer #2: Yes

3. Has the statistical analysis been performed appropriately and rigorously? 

Reviewer #1: Yes

Reviewer #2: Yes

4. Have the authors made all data underlying the findings in their manuscript fully available?

Reviewer #1: Yes

Reviewer #2: Yes

5. Is the manuscript presented in an intelligible fashion and written in standard English?

Reviewer #1: Yes

Reviewer #2: Yes

6. Review Comments to the Author

Reviewer #1: (No Response)

Reviewer #2: Chang et al. conducted a systematic review and meta-analysis on “Impact of PARP inhibitor maintenance therapy in newly diagnosed advanced epithelial ovarian cancer (EOC)”. They concluded that in newly diagnosed advanced EOC patients, PARPi maintenance therapy was significantly more effective in terms of survival compared to placebo group. The authors have made significant improvements in this revised manuscript while the issue of small sample size is still in its place. That being said, this study can potentially provide insights into the existing scientific knowledge pertaining to the therapeutic strategies of EOCs and is therefore recommended for publication in PLOS ONE.

7. PLOS authors have the option to publish the peer review history of their article (what does this mean?). If published, this will include your full peer review and any attached files.

Reviewer #1: No

Reviewer #2: **Yes: **Kidane Siele Embaye

---

## [Editor Report · Acceptance letter]

9 Nov 2023

PONE-D-23-26174R1 

Impact of PARP inhibitor maintenance therapy in newly diagnosed advanced epithelial ovarian cancer: A meta-analysis 

Dear Dr. Chang:

I'm pleased to inform you that your manuscript has been deemed suitable for publication in PLOS ONE. Congratulations! Your manuscript is now with our production department. 

Kind regards, 

on behalf of

Dr. Asmerom Tesfamariam Sengal 

Academic Editor

PLOS ONE